# Breaking the Sample Size Barrier in Model-Based Reinforcement Learning with a Generative Model

| Gen Li | Yuting Wei | Yuejie Chi | Yuantao Gu | Yuxin Chen |
|--------|-----------|-----------|-----------|-----------|
| Tsinghua | CMU | CMU | Tsinghua | Princeton |

## Abstract

We investigate the sample efficiency of reinforcement learning in a $\gamma$-discounted infinite-horizon Markov decision process (MDP) with state space $\mathcal{S}$ and action space $\mathcal{A}$, assuming access to a generative model. Despite a number of prior work tackling this problem, a complete picture of the trade-offs between sample complexity and statistical accuracy is yet to be determined. In particular, prior results suffer from a sample size barrier, in the sense that their claimed statistical guarantees hold only when the sample size exceeds at least $\frac{|\mathcal{S}||\mathcal{A}|}{(1-\gamma)^2}$ (up to some log factor). The current paper overcomes this barrier by certifying the minimax optimality of model-based reinforcement learning as soon as the sample size exceeds the order of $\frac{|\mathcal{S}||\mathcal{A}|}{1-\gamma}$ (modulo some log factor). More specifically, a *perturbed* model-based planning algorithm provably finds an $\varepsilon$-optimal policy with an order of $\frac{|\mathcal{S}||\mathcal{A}|}{(1-\gamma)^3\varepsilon^2} \log \frac{|\mathcal{S}||\mathcal{A}|}{(1-\gamma)\varepsilon}$ samples for any $\varepsilon \in (0, \frac{1}{1-\gamma}]$. Along the way, we derive improved (instance-dependent) guarantees for model-based policy evaluation. To the best of our knowledge, this work provides the first minimax-optimal guarantee in a generative model that accommodates the entire range of sample sizes (beyond which finding a meaningful policy is information theoretically impossible).

## 1 Introduction

Reinforcement learning (RL) [33, 35], which is frequently modeled as learning and decision making in a Markov decision process (MDP), is garnering growing interest in recent years due to its remarkable success in practice. A core objective of RL is to search for a policy — based on a collection of noisy data samples — that approximately maximizes expected rewards in an MDP, without direct access to a precise description of the underlying model.[1] To enable reliable RL in the sample-starved regime (i.e. the regime where the model complexity overwhelms the sample size), it is crucial to obtain a quantitative picture of the trade-off between sample complexity and statistical accuracy, and to design efficient algorithms that provably achieve the optimal trade-off. Characterizing the sample complexities of RL algorithms (including both model-based and model-free approaches) has been the focal point of a large body of recent work, e.g. [1, 2, 15, 19, 29, 30, 38, 39, 40, 41].

This paper pursues a comprehensive understanding about the sample efficiency of model-based RL, assuming access to a generative model — that is, a simulator that produces samples based on the transition kernel of the MDP for each state-action pair [16, 19]. We focus on an infinite-horizon discounted MDP with state space $\mathcal{S}$, action space $\mathcal{A}$, and discount factor $0 < \gamma < 1$. We obtain $N$ samples per state-action pair by querying the generative model. For an *arbitrary* target accuracy level $\varepsilon > 0$, the goal is to compute an $\varepsilon$-optimal policy with a minimal number of calls to the generative model. Particular emphasis is placed on the sub-linear sampling scenario, in which the total sample size is smaller than the total number $|\mathcal{S}|^2|\mathcal{A}|$ of model parameters (so that it might be infeasible to estimate the model accurately).

| Algorithm | Sample size range | Sample complexity | $\varepsilon$-range |
|---|---|---|---|
| Phased Q-learning [19] | $\left[\frac{\|\mathcal{S}\|\|\mathcal{A}\|}{(1-\gamma)^5}, \infty\right)$ | $\frac{\|\mathcal{S}\|\|\mathcal{A}\|}{(1-\gamma)^7\varepsilon^2}$ | $\left(0, \frac{1}{1-\gamma}\right]$ |
| Empirical QVI [2] | $\left[\frac{\|\mathcal{S}\|^2\|\mathcal{A}\|}{(1-\gamma)^2}, \infty\right)$ | $\frac{\|\mathcal{S}\|\|\mathcal{A}\|}{(1-\gamma)^3\varepsilon^2}$ | $\left(0, \frac{1}{\sqrt{(1-\gamma)\|\mathcal{S}\|}}\right]$ |
| Sublinear randomized VI [30] | $\left[\frac{\|\mathcal{S}\|\|\mathcal{A}\|}{(1-\gamma)^2}, \infty\right)$ | $\frac{\|\mathcal{S}\|\|\mathcal{A}\|}{(1-\gamma)^4\varepsilon^2}$ | $\left(0, \frac{1}{1-\gamma}\right]$ |
| Variance-reduced QVI [29] | $\left[\frac{\|\mathcal{S}\|\|\mathcal{A}\|}{(1-\gamma)^3}, \infty\right)$ | $\frac{\|\mathcal{S}\|\|\mathcal{A}\|}{(1-\gamma)^3\varepsilon^2}$ | $(0, 1]$ |
| Randomized primal-dual alg. [41] | $\left[\frac{\|\mathcal{S}\|\|\mathcal{A}\|}{(1-\gamma)^2}, \infty\right)$ | $\frac{\|\mathcal{S}\|\|\mathcal{A}\|}{(1-\gamma)^4\varepsilon^2}$ | $\left(0, \frac{1}{1-\gamma}\right]$ |
| Empirical MDP + planning [1] | $\left[\frac{\|\mathcal{S}\|\|\mathcal{A}\|}{(1-\gamma)^2}, \infty\right)$ | $\frac{\|\mathcal{S}\|\|\mathcal{A}\|}{(1-\gamma)^3\varepsilon^2}$ | $\left(0, \frac{1}{\sqrt{1-\gamma}}\right]$ |
| *Perturbed* empirical MDP + planning **This paper** | $\left[\frac{\|\mathcal{S}\|\|\mathcal{A}\|}{1-\gamma}, \infty\right)$ | $\frac{\|\mathcal{S}\|\|\mathcal{A}\|}{(1-\gamma)^3\varepsilon^2}$ | $\left(0, \frac{1}{1-\gamma}\right]$ |

Table 1: Comparisons with prior results (up to log factors) regarding finding an $\varepsilon$-optimal policy with a generative model. The sample size range and the $\varepsilon$-range stand for the range of sample size and optimality gap (e.g. $\varepsilon$-accuracy) for the claimed sample complexity to hold. Note that the results in [19, 41] only hold for a restricted family of MDPs satisfying certain ergodicity assumptions.

**Motivation: sample size barriers.** Several prior work was dedicated to investigating model-based RL with a generative model, which uncovered the minimax optimality of this approach for an already wide regime [1, 2]. However, their results often suffered from a sample size barrier that prevents us from obtaining a complete trade-off curve between sample size and accuracy. For instance, the state-of-the-art result [1] required the total sample size to at least exceed $\frac{\|\mathcal{S}\|\|\mathcal{A}\|}{(1-\gamma)^2}$ (up to some log factor), thus restricting the validity of the theory for broader contexts. In truth, this is not merely an issue for model-based planning; the same barrier already showed up when analyzing the simpler task of model-based policy evaluation [1, 26]. Furthermore, an even higher barrier emerged in prior theory for model-free methods; for instance, [30, 40] required the sample size to exceed $\frac{\|\mathcal{S}\|\|\mathcal{A}\|}{(1-\gamma)^3}$ (up to some log factor). In stark contrast, however, no lower bounds developed thus far preclude us from attaining reasonable statistical accuracy when going below such sample size barriers.

**Our contributions.** Following the model-based approach, we propose to perform planning based on an empirical MDP learned from samples with mild *reward perturbation*. The perturbed model-based algorithm we propose finds an $\varepsilon$-optimal policy with an order of $\frac{\|\mathcal{S}\|\|\mathcal{A}\|}{(1-\gamma)^3\varepsilon^2}$ samples (up to log factor), which matches the minimax lower bound [2]. Our result accommodates the full range of accuracy level $\varepsilon$ (i.e. $\varepsilon \in (0, \frac{1}{1-\gamma}]$), thus unveiling the minimaxity of our algorithm as soon as the sample size exceeds $\frac{\|\mathcal{S}\|\|\mathcal{A}\|}{1-\gamma}$ (modulo some log factor); encouragingly, this covers the *full* range of sample sizes that enable one to find a policy strictly better than a random guess. See Table 1 for detailed comparisons with prior work. Along the way, we also derive (instance-dependent) guarantees for policy evaluation, which strengthens state-of-the-art results by broadening the sample size range.

Our theory is established upon a novel combination of several key ideas: (1) a high-order expansion of the value function estimation error, coupled with fine-grained analysis for each term in the expansion; (2) the construction of auxiliary leave-one-out type (state-action-absorbing) MDPs — motivated by [1] — that help decouple the complicated statistical dependency between the empirically optimal policy (as opposed to value functions) and data samples; (3) a tie-breaking argument guaranteeing that the empirically optimal policy stands out from all other policies under reward perturbations.

## 2 Problem formulation

**Basics of Markov decision processes.** Consider an infinite-horizon discounted MDP represented by a quintuple $\mathcal{M} = (\mathcal{S}, \mathcal{A}, P, r, \gamma)$, where $\mathcal{S} := \{1, 2, \ldots, \|\mathcal{S}\|\}$ denotes a finite set of states, $\mathcal{A} := \{1, 2, \ldots, \|\mathcal{A}\|\}$ is a finite set of actions, $\gamma \in (0, 1)$ is the discount factor, and $r : \mathcal{S} \times \mathcal{A} \to [0, 1]$ represents the reward function, i.e. $r(s, a)$ is the immediate reward received upon executing action $a$ while in state $s$ (here and throughout, we consider the normalized setting where the rewards lie within $[0, 1]$). In addition, $P : \mathcal{S} \times \mathcal{A} \to \Delta(\mathcal{S})$ represents the probability transition kernel, where

$P(s'|s, a)$ denotes the probability of transiting from state $s$ to state $s'$ when action $a$ is executed, and $\Delta(\mathcal{S})$ denotes the probability simplex over $\mathcal{S}$.

A deterministic policy is a mapping $\pi : \mathcal{S} \to \mathcal{A}$, which maps a state to an action. The value function $V^\pi : \mathcal{S} \to \mathbb{R}$ of a policy $\pi$ is defined by

$$\forall s \in \mathcal{S} : \qquad V^\pi(s) := \mathbb{E}\left[\sum\nolimits_{t=0}^\infty \gamma^t r(s^t, a^t) \,\big|\, s^0 = s\right], \tag{1}$$

which is the expected discounted total reward starting from the initial state $s^0 = s$, with the actions taken according to the policy $\pi$ (namely, $a^t = \pi(s^t)$ for all $t \geq 0$) and the trajectory generated based on the transition kernel (namely, $s^{t+1} \sim P(\cdot|s^t, a^t)$). It is easily seen that $0 \leq V^\pi(s) \leq \frac{1}{1-\gamma}$. The corresponding action-value function (or Q-function) $Q^\pi : \mathcal{S} \times \mathcal{A} \to \mathbb{R}$ of a policy $\pi$ is defined by

$$\forall (s, a) \in \mathcal{S} \times \mathcal{A} : \qquad Q^\pi(s, a) := \mathbb{E}\left[\sum\nolimits_{t=0}^\infty \gamma^t r(s^t, a^t) \,\big|\, s^0 = s, a^0 = a\right], \tag{2}$$

where the actions are taken according to the policy $\pi$ after the initial action (i.e. $a^t = \pi(s^t)$ for all $t \geq 1$). It is known that there exists an optimal policy, denoted by $\pi^\star$, that maximizes $V^\pi(s)$ (resp. $Q^\pi(s, a)$) for all $s \in \mathcal{S}$ (resp. $(s, a) \in (\mathcal{S} \times \mathcal{A})$) [33]. The corresponding value function $V^\star := V^{\pi^\star}$ (resp. Q-function $Q^\star := Q^{\pi^\star}$) is called the optimal value function (resp. Q-function).

**A generative model and an empirical MDP.** The current paper focuses on a stylized generative model (also called a simulator) as studied in [16, 18]. Assuming access to this generative model, we collect $N$ independent samples $s_{s,a}^i \sim P(\cdot|s, a)$, $i = 1, \ldots, N$ for any state-action pair $(s, a)$, which allows us to construct an empirical transition kernel $\widehat{P}$ given by

$$\forall s' \in \mathcal{S}, \qquad \widehat{P}(s'|s, a) = \frac{1}{N}\sum\nolimits_{i=1}^N \mathbb{1}\{s_{s,a}^i = s'\}, \tag{3}$$

where $\mathbb{1}\{\cdot\}$ is the indicator function. In words, $\widehat{P}(s'|s, a)$ counts the empirical frequency of transitions from $(s, a)$ to state $s'$. The total sample size should be understood as $N^{\mathsf{total}} := N|\mathcal{S}||\mathcal{A}|$. This leads to an empirical MDP $\widehat{\mathcal{M}} = (\mathcal{S}, \mathcal{A}, \widehat{P}, r, \gamma)$ constructed from the data samples. We can define the value function and the action-value function of a policy $\pi$ for $\widehat{\mathcal{M}}$ analogously, which we shall denote by $\widehat{V}^\pi$ and $\widehat{Q}^\pi$, respectively. The optimal policy of $\widehat{\mathcal{M}}$ is denoted by $\widehat{\pi}^\star$, with the optimal value function and Q-function denoted by $\widehat{V}^\star := \widehat{V}^{\widehat{\pi}^\star}$ and $\widehat{Q}^\star := \widehat{Q}^{\widehat{\pi}^\star}$, respectively.

**Policy evaluation and planning.** Given a few data samples in hand, the task of policy evaluation aims to compute or approximate the value function $V^\pi$ under this policy. To be precise, for any target level $\varepsilon > 0$, the goal is to find an $\varepsilon$-accurate estimate $V_{\mathsf{est}}^\pi$ such that

$$\text{(policy evaluation)} \qquad \forall s \in \mathcal{S} : \qquad \left|V_{\mathsf{est}}^\pi(s) - V^\pi(s)\right| \leq \varepsilon. \tag{4}$$

In contrast, the task of planning seeks a policy that (approximately) maximizes the expected discounted reward. Specifically, for any target $\varepsilon > 0$, we aim to find an $\varepsilon$-optimal policy $\pi_{\mathsf{est}}$ obeying

$$\text{(planning)} \qquad \forall (s, a) \in \mathcal{S} \times \mathcal{A} : \qquad V^{\pi_{\mathsf{est}}}(s) \geq V^\star(s) - \varepsilon, \ Q^{\pi_{\mathsf{est}}}(s, a) \geq Q^\star(s, a) - \varepsilon. \tag{5}$$

Naturally, one would hope to accomplish these tasks with as few samples as possible. Recall that for the normalized reward setting ($0 \leq r \leq 1$), the value function and Q-function fall in the range $[0, \frac{1}{1-\gamma}]$; this means that the range of accuracy level $\varepsilon$ should be set to $\varepsilon \in [0, \frac{1}{1-\gamma}]$. A model-based approach starts by constructing an empirical MDP $\widehat{\mathcal{M}}$ based on all collected samples, and then "plugs in" this empirical model directly into the Bellman recursion [5] to perform policy evaluation or planning, with prominent examples including Q-value iteration (QVI) and policy iteration (PI).

**Notation** Let $\mathcal{X} := \left(|\mathcal{S}|, |\mathcal{A}|, \frac{1}{1-\gamma}, \frac{1}{\varepsilon}\right)$. The notation $f(\mathcal{X}) = O(g(\mathcal{X}))$ means that there exists a constant $C_1 > 0$ such that $f \leq C_1 g$, whereas $f(\mathcal{X}) = \Omega(g(\mathcal{X}))$ means $g(\mathcal{X}) = O(f(\mathcal{X}))$. The notation $\widetilde{O}(\cdot)$ (resp. $\widetilde{\Omega}(\cdot)$) is defined in the same way as $O(\cdot)$ (resp. $\Omega(\cdot)$) except that it ignores log factors. For any vector $\boldsymbol{a} = [a_i]_{1 \leq i \leq n} \in \mathbb{R}^n$, we overload the notation $\sqrt{\cdot}$ and $|\cdot|$ in an entry-wise manner s.t. $\sqrt{\boldsymbol{a}} := [\sqrt{a_i}]_{1 \leq i \leq n}$ and $|\boldsymbol{a}| := [|a_i|]_{1 \leq i \leq n}$. For any vectors $\boldsymbol{a} = [a_i]_{1 \leq i \leq n}$ and $\boldsymbol{b} = [b_i]_{1 \leq i \leq n}$, the notation $\boldsymbol{a} \geq \boldsymbol{b}$ (resp. $\boldsymbol{a} \leq \boldsymbol{b}$) means $a_i \geq b_i$ (resp. $a_i \leq b_i$) for all $1 \leq i \leq n$, and we let $\boldsymbol{a} \circ \boldsymbol{b} := [a_i b_i]_{1 \leq i \leq n}$ represent the Hadamard product. We denote by $\boldsymbol{1}$ the all-one vector.

# 3 Main results

To break the aforementioned sample size barrier, we propose to invoke the model-based planning approach applied to an empirical MDP with *perturbed rewards*. Specifically, for each state-action pair $(s, a) \in \mathcal{S} \times \mathcal{A}$, let us perturb the immediate reward (with uniform distribution) to obtain

$$r_{\mathrm{p}}(s, a) = r(s, a) + \zeta(s, a), \qquad \zeta(s, a) \overset{\text{i.i.d.}}{\sim} \mathsf{Unif}(0, \xi), \tag{6}$$

where $\xi > 0$ is a parameter to be specified shortly.[2] For any policy $\pi$, denote by $\widehat{V}_{\mathrm{p}}^{\pi}$ the corresponding value function of the perturbed empirical MDP $\widehat{\mathcal{M}}_{\mathrm{p}} = (\mathcal{S}, \mathcal{A}, \widehat{P}, r_{\mathrm{p}}, \gamma)$ with the probability transition kernel $\widehat{P}$ and the perturbed reward function $r_{\mathrm{p}}$. Let $\widehat{\pi}_{\mathrm{p}}^{\star}$ represent the optimal policy of $\widehat{\mathcal{M}}_{\mathrm{p}}$, i.e.

$$\widehat{\pi}_{\mathrm{p}}^{\star} := \arg\max_{\pi} \widehat{V}_{\mathrm{p}}^{\pi}. \tag{7}$$

Encouragingly, this policy results in a value function $V^{\widehat{\pi}_{\mathrm{p}}^{\star}}$ (resp. Q-function $Q^{\widehat{\pi}_{\mathrm{p}}^{\star}}$) that is $\varepsilon$-close to the true optimal value function $V^{\star}$ (resp. optimal Q-function $Q^{\star}$), as asserted below.

**Theorem 1** (Perturbed model-based planning)**.** *There exist some universal constants $c_0, c_1 > 0$ such that: for any $\delta > 0$ and any $0 < \varepsilon \leq \frac{1}{1-\gamma}$, with probability at least $1 - \delta$ the policy $\widehat{\pi}_{\mathrm{p}}^{\star}$ (cf. (7)) obeys*

$$\forall (s, a) \in \mathcal{S} \times \mathcal{A}, \qquad V^{\widehat{\pi}_{\mathrm{p}}^{\star}}(s) \geq V^{\star}(s) - \varepsilon \qquad \text{and} \qquad Q^{\widehat{\pi}_{\mathrm{p}}^{\star}}(s, a) \geq Q^{\star}(s, a) - \gamma\varepsilon, \tag{8}$$

*provided that the perturbation obeys $\xi = \frac{c_1(1-\gamma)\varepsilon}{|\mathcal{S}|^5|\mathcal{A}|^5}$ and the sample size per state-action pair obeys*

$$N \geq \frac{c_0 \log\left(\frac{|\mathcal{S}||\mathcal{A}|}{(1-\gamma)\varepsilon\delta}\right)}{(1-\gamma)^3 \varepsilon^2}. \tag{9}$$

*In addition, the empirical QVI (cf. [2]) recovers $\widehat{\pi}_{\mathrm{p}}^{\star}$ perfectly within $O\left(\frac{1}{1-\gamma} \log\left(\frac{|\mathcal{S}||\mathcal{A}|}{(1-\gamma)\varepsilon\delta}\right)\right)$ iterations.*

***Remark* 1.** Theorem 1 holds unchanged if $\xi$ is taken to be $\frac{c_1(1-\gamma)\varepsilon}{|\mathcal{S}|^{\alpha}|\mathcal{A}|^{\alpha}}$ for any $\alpha \geq 1$. The current paper picks the specific choice $\alpha = 5$ merely to convey that a very small level of perturbation suffices.

***Remark* 2.** Perturbation brings a side benefit: one can recover the optimal policy $\widehat{\pi}_{\mathrm{p}}^{\star}$ of the perturbed empirical MDP $\widehat{\mathcal{M}}_{\mathrm{p}}$ exactly in a few iterations without incurring further optimization errors. To give a flavor of the overall computational cost, recall that each iteration of QVI takes time proportional to the time taken to read $\widehat{P}$ (which is a sparse matrix with at most $N|\mathcal{S}||\mathcal{A}|$ nonzeros), hence the resulting computational complexity can be as low as $O\left(\frac{|\mathcal{S}||\mathcal{A}|}{(1-\gamma)^4 \varepsilon^2} \log^2\left(\frac{|\mathcal{S}||\mathcal{A}|}{(1-\gamma)\varepsilon\delta}\right)\right)$.

Theorem 1 demonstrates that: the proposed perturbed model-based approach finds an $\varepsilon$-optimal policy as soon as the total sample complexity exceeds the order of $\frac{|\mathcal{S}||\mathcal{A}|}{(1-\gamma)^3 \varepsilon^2}$ (modulo some log factor). Compared to prior literature, our result imposes no restriction on the range of $\varepsilon$ and, in particular, we allow the accuracy level $\varepsilon$ to go all the way up to $\frac{1}{1-\gamma}$. Our result is particularly useful in the regime with small-to-moderate sample sizes, since its validity is guaranteed as long as

$$N \geq \widetilde{\Omega}\left(\frac{1}{1-\gamma}\right). \tag{10}$$

Tackling the sample-limited regime (particularly the scenario when $N \in [\frac{1}{1-\gamma}, \frac{1}{(1-\gamma)^2}]$) requires us to develop new analysis ideas beyond prior theory, which we shall discuss in detail momentarily.

We remark that [2] established a minimax lower bound of the same order as (9) (up to log factor) in the regime $\varepsilon = O(1)$. A closer inspection of their analysis, however, reveals that their argument holds true as long as $\varepsilon = O(\frac{1}{1-\gamma})$. This in turn corroborates the *minimax optimality* of our perturbed model-based approach for the full $\varepsilon$-range (which is previously unavailable), and demonstrates the information-theoretic infeasibility to learn a policy strictly better than a random guess if $N \leq \widetilde{O}\left(\frac{1}{1-\gamma}\right)$. Put another way, the condition (10) contains the full range of "meaningful" sample sizes.

To further demonstrate the effectiveness of our model-based approach, we conduct a series of numerical experiments (motivated by [2]), focusing on the effect of the discount complexity $\frac{1}{1-\gamma}$

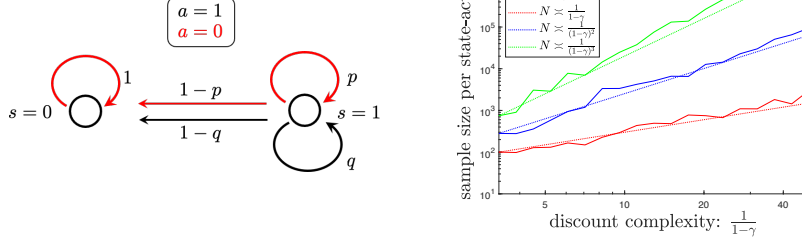

Figure 1: Numerical sample complexity per state-action pair $N$ vs. discount complexity $\frac{1}{1-\gamma}$.

upon the sample complexity. More specifically, consider the following MDP $(\mathcal{S}, \mathcal{A}, P, r, \gamma)$, where $\mathcal{S} = \{0, 1\}$, $\mathcal{A}_0 = \{0\}$, $\mathcal{A}_1 = \{0, 1\}$, $P(0|0, 0) = 1, P(1|1, 0) = p, P(1|1, 1) = q$ and $r(0, 0) = 0, r(1, 0) = r(1, 1) = 1$ (see Fig. 1 for an illustration), and we take the quantities $p$ and $q$ to be $p = \gamma + \frac{2\gamma(1-\gamma)^2\varepsilon}{(1+\gamma)^2}$ and $q = \gamma - \frac{2\gamma(1-\gamma)^2\varepsilon}{(1+\gamma)^2}$ as suggested by [2]. For each point $(N, \frac{1}{1-\gamma}, \varepsilon)$, we perform 100 Monte Carlo simulations, and we declare $N$ to successfully attain $\varepsilon$ accuracy if this accuracy is achieved in at least 95% of the trials. In Fig. 1, the solid lines depict the minimum sample size per state-action pair needed to achieve $\varepsilon$ accuracy numerically as the discount complexity varies. As can be seen, the numerical sample complexity per state-action pair scales on the order of $\frac{1}{(1-\gamma)^3\varepsilon^2}$ for varying choices of $\varepsilon$, which is consistent with our theory.

Finally, we single out an intermediate result in the analysis of Theorem 1 concerning model-based policy evaluation, which might be of interest on its own. Specifically, for any fixed policy $\pi$, this task concerns value function estimation via the plug-in estimate $\widehat{V}^\pi$ (i.e. the value function of the empirical $\mathcal{M}$ under this policy). However simple as this might seem, existing theoretical underpinnings of this approach remain suboptimal, unless the sample size is already large enough. Our result is this:

**Theorem 2** (Model-based policy evaluation). *Fix any policy $\pi$. There exists some universal constant $c_0 > 0$ such that: for any $0 < \delta < 1$ and $0 < \varepsilon \leq \frac{1}{1-\gamma}$, one has*

$$\max_{s \in \mathcal{S}} \left| \widehat{V}^\pi(s) - V^\pi(s) \right| \leq \varepsilon \tag{11}$$

*with probability at least $1 - \delta$, provided that the sample size per state-action pair exceeds*

$$N \geq c_0 \frac{\log\left(\frac{|\mathcal{S}| \log \frac{e}{1-\gamma}}{\delta}\right)}{(1-\gamma)^3\varepsilon^2}. \tag{12}$$

In words, this theorem reveals that $\widehat{V}^\pi$ begins to outperform a random guess as soon as $N \geq \widetilde{\Omega}\left(\frac{1}{1-\gamma}\right)$. The sample complexity bound (12) enjoys *full* coverage of the $\varepsilon$-range $(0, \frac{1}{1-\gamma}]$, and matches the minimax lower bound [26, Theorem 2(b)] up to only a $\log\log\frac{1}{1-\gamma}$ factor. In addition, a recent line of work investigated instance-dependent guarantees for policy evaluation ([20, 26]). While this is not our focus, our analysis does uncover an improved instance-dependent bound. See Lemma 1.

**Prior bounds for planning and policy learning.** None of the prior results with a generative model (including both model-based and model-free methods) was capable of efficiently finding the desired policy while accommodating the full sample size range (10). For instance, the state-of-the-art analysis for the model-based approach [1] required the sample size to exceed $N \geq \widetilde{\Omega}\left(\frac{1}{(1-\gamma)^2}\right)$, whereas the theory for the variance-reduced model-free approach [29, 40] imposed the sample size requirement $N \geq \widetilde{\Omega}\left(\frac{1}{(1-\gamma)^3}\right)$. In fact, it was previously unknown what is achievable in the sample size range $N \in [\frac{1}{1-\gamma}, \frac{1}{(1-\gamma)^2}]$. In contrast, our results confirm the minimax-optimal statistical performance of the model-based approach with full coverage of the $\varepsilon$-range and the sample size range.

**Prior bounds for policy evaluation.** Prior results [1, 2, 26] for the plug-in approach operated under the assumption that $N \geq \widetilde{\Omega}\left(\frac{1}{(1-\gamma)^2}\right)$, which is more stringent than our result by a factor of at least

$\frac{1}{1-\gamma}$. Our sample complexity matches the state-of-the-art guarantees in the regime where $\varepsilon \leq \frac{1}{\sqrt{1-\gamma}}$ [1, 26], while extending them to the range $\varepsilon \in \left[\frac{1}{\sqrt{1-\gamma}}, \frac{1}{1-\gamma}\right]$ uncovered previously.

**Other related work.** Classical analyses of reinforcement learning algorithms have largely focused on asymptotic performance (e.g. [14, 34, 36, 37]). Leveraging the toolkit of concentration inequalities, a number of recent papers have shifted attention towards understanding the performance in the non-asymptotic and finite-time settings. A highly incomplete list includes [3, 4, 6, 7, 8, 9, 10, 11, 12, 13, 15, 17, 19, 20, 21, 24, 25, 27, 28, 29, 31, 32, 40, 42, 43, 44], a large fraction of which is concerned with model-free algorithms.

The generative model adopted herein was first proposed in [19], which has been invoked in [1, 2, 16, 18, 19, 20, 22, 26, 29, 30, 40, 41, 45], to name a few. In particular, [2] developed the minimax lower bound on the sample complexity $N = \Omega\left(\frac{|\mathcal{S}||\mathcal{A}| \log(|\mathcal{S}||\mathcal{A}|)}{(1-\gamma)^3 \varepsilon^2}\right)$ necessary for finding an $\varepsilon$-optimal policy, and showed that, for any $\varepsilon \in (0, 1)$, a model-based approach (e.g. applying QVI to the empirical MDP) can estimate the optimal Q-function to within an $\varepsilon$-accuracy given near-minimal samples. Note, however, that directly translating this result to the policy guarantees leads to an extra factor of $\frac{1}{1-\gamma}$ in estimation accuracy and of $\frac{1}{(1-\gamma)^2}$ in sample complexity. In light of this, [2] further showed that a near-optimal sample complexity is possible for policy learning if the sample size is at least on the order of $\frac{|\mathcal{S}|^2|\mathcal{A}|}{(1-\gamma)^2}$ which, however, is no longer sub-linear in the model complexity. A recent breakthrough [1] substantially improved the model-based guarantee with the aid of auxiliary state-absorbing MDPs, extending the sample complexity range to $\left[\frac{|\mathcal{S}||\mathcal{A}| \log(|\mathcal{S}||\mathcal{A}|)}{(1-\gamma)^2}, \infty\right)$. Our analysis is motivated in part by [1], but also relies on several other novel techniques to complete the picture.

## 4 Analysis

This section sketches the key proof ideas, with full details postponed to the full version [23].

**Matrix notation.** We first introduce a few matrix notation. Denoting by $\boldsymbol{e}_1, \cdots, \boldsymbol{e}_{|\mathcal{S}|} \in \mathbb{R}^{|\mathcal{S}|}$ the standard basis vectors, we can define $\boldsymbol{r} \in \mathbb{R}^{|\mathcal{S}||\mathcal{A}|}$, $\boldsymbol{V}^\pi \in \mathbb{R}^{|\mathcal{S}|}$, $\boldsymbol{Q}^\pi \in \mathbb{R}^{|\mathcal{S}||\mathcal{A}|}$ as the vector representations of the reward function $r$, the value function $V^\pi$ and the Q-function $Q^\pi$. Let $\boldsymbol{V}^\star$ and $\boldsymbol{Q}^\star \in \mathbb{R}^{|\mathcal{S}||\mathcal{A}|}$ be the optimal value function $V^\star$ and optimal Q-function $Q^\star$. Let $\boldsymbol{P} \in \mathbb{R}^{|\mathcal{S}||\mathcal{A}| \times |\mathcal{S}|}$ be a matrix representing the probability transition kernel $P$, where the $(s, a)$-th row of $\boldsymbol{P}$ is a probability vector representing $P(\cdot|s, a)$.

We introduce $\boldsymbol{\Pi}^\pi = \begin{pmatrix} \boldsymbol{e}_{\pi(1)}^\top & & & \\ & \boldsymbol{e}_{\pi(2)}^\top & & \\ & & \ddots & \\ & & & \boldsymbol{e}_{\pi(|\mathcal{S}|)}^\top \end{pmatrix} \in \{0, 1\}^{|\mathcal{S}| \times |\mathcal{S}||\mathcal{A}|}$ as the projection matrix associated with policy $\pi$. The matrices $\boldsymbol{P}^\pi := \boldsymbol{P}\boldsymbol{\Pi}^\pi \in \mathbb{R}^{|\mathcal{S}||\mathcal{A}| \times |\mathcal{S}||\mathcal{A}|}$ and $\boldsymbol{P}_\pi := \boldsymbol{\Pi}^\pi \boldsymbol{P} \in \mathbb{R}^{|\mathcal{S}| \times |\mathcal{S}|}$ are two *square* probability transition matrices induced by policy $\pi$ over the state-action pairs and the states, respectively. Let $\boldsymbol{r}_\pi \in \mathbb{R}^{|\mathcal{S}|}$ be a reward vector restricted to the actions chosen by policy $\pi$, i.e. $\boldsymbol{r}_\pi = \boldsymbol{\Pi}^\pi \boldsymbol{r}$. For a vector $\boldsymbol{V} = [V_i]_{1 \leq i \leq |\mathcal{S}|} \in \mathbb{R}^{|\mathcal{S}|}$, we define the vectors

$$\mathsf{Var}_{\boldsymbol{P}}(\boldsymbol{V}) := \boldsymbol{P}(\boldsymbol{V} \circ \boldsymbol{V}) - (\boldsymbol{P}\boldsymbol{V}) \circ (\boldsymbol{P}\boldsymbol{V}) \in \mathbb{R}^{|\mathcal{S}||\mathcal{A}|}; \quad \mathsf{Var}_{\boldsymbol{P}_\pi}(\boldsymbol{V}) := \boldsymbol{\Pi}^\pi \mathsf{Var}_{\boldsymbol{P}}(\boldsymbol{V}) \in \mathbb{R}^{|\mathcal{S}|}.$$

We shall also define $\widehat{\boldsymbol{V}}^\pi$, $\widehat{\boldsymbol{Q}}^\pi$, $\widehat{\boldsymbol{V}}^\star$, $\widehat{\boldsymbol{Q}}^\star$, $\widehat{\boldsymbol{P}}$, $\widehat{\boldsymbol{P}}^\pi$, $\widehat{\boldsymbol{P}}_\pi$, $\mathsf{Var}_{\widehat{\boldsymbol{P}}}(\boldsymbol{V})$, $\mathsf{Var}_{\widehat{\boldsymbol{P}}_\pi}(\boldsymbol{V})$ w.r.t. $\widehat{\mathcal{M}}$ analogously.

### 4.1 Key ideas of the proofs

**Analysis: model-based policy evaluation.** We start with the simpler task of policy evaluation, which is crucial in subsequent analysis. To establish Theorem 2, we aim to prove the following result. In view of the Bellman equation [5] and with our matrix notation in place, the true value function under policy $\pi$ and the model-based empirical estimate can be written respectively as

$$\boldsymbol{V}^\pi = (\boldsymbol{I} - \gamma \boldsymbol{P}_\pi)^{-1} \boldsymbol{r}_\pi \qquad \text{and} \qquad \widehat{\boldsymbol{V}}^\pi = (\boldsymbol{I} - \gamma \widehat{\boldsymbol{P}}_\pi)^{-1} \boldsymbol{r}_\pi. \tag{13}$$

**Lemma 1.** *Fix any policy $\pi$. Consider any $0 < \delta < 1$, and suppose $N \geq \frac{32e^2}{1-\gamma} \log\left(\frac{4|\mathcal{S}|\log(\frac{e}{1-\gamma})}{\delta}\right)$. Then with probability at least $1 - \delta$, the vectors defined in* (13) *obey*

$$\left\|\widehat{\boldsymbol{V}}^{\pi} - \boldsymbol{V}^{\pi}\right\|_{\infty} \leq \gamma\sqrt{\frac{32\log\frac{4|\mathcal{S}|\log(\frac{e}{1-\gamma})}{\delta}}{N}}\left\|(\boldsymbol{I} - \gamma\boldsymbol{P}_{\pi})^{-1}\sqrt{\mathsf{Var}_{\boldsymbol{P}_{\pi}}[\boldsymbol{V}^{\pi}]}\right\|_{\infty} + \frac{2\gamma\log\frac{4|\mathcal{S}|\log(\frac{e}{1-\gamma})}{\delta}}{(1-\gamma)N}\left\|\boldsymbol{V}^{\pi}\right\|_{\infty}$$

$$\leq 6\sqrt{\frac{2\log\left(\frac{4|\mathcal{S}|\log(\frac{e}{1-\gamma})}{\delta}\right)}{N(1-\gamma)^3}}. \tag{14}$$

Here, we strengthen Theorem 2 by providing an additional instance-dependent bound (see the first line of (14) that depends on the true instance $\boldsymbol{P}_{\pi}, \boldsymbol{V}^{\pi}$), which is often tighter and more adaptive than the worst-case bound stated in the second line of (14). Our contribution can be better understood when compared with [26]. Assuming that there is no noise in the rewards, our instance-dependent guarantee matches [26, Theorem 1(a)] up to some $\log\log\frac{1}{1-\gamma}$ factor, while being capable of covering the full sample size range $N \geq \widetilde{\Omega}(\frac{1}{1-\gamma})$. In contrast, [26, Theorem 1] is only valid when $N \geq \widetilde{\Omega}(\frac{1}{(1-\gamma)^2})$.

The key proof idea is to resort to a high-order successive expansion of $\widehat{\boldsymbol{V}}^{\pi} - \boldsymbol{V}^{\pi}$, followed by fine-grained analysis of each term up to a certain logarithmic order. To catch a glimpse of the motivation, we begin with the following elementary, and commonly used, perterbation relation

$$\widehat{\boldsymbol{V}}^{\pi} - \boldsymbol{V}^{\pi} = \left(\boldsymbol{I} - \gamma\widehat{\boldsymbol{P}}_{\pi}\right)^{-1}\boldsymbol{r}_{\pi} - \boldsymbol{V}^{\pi} = \gamma\left(\boldsymbol{I} - \gamma\widehat{\boldsymbol{P}}_{\pi}\right)^{-1}\left(\widehat{\boldsymbol{P}}_{\pi} - \boldsymbol{P}_{\pi}\right)\boldsymbol{V}^{\pi}. \tag{15}$$

Due to the complicated statistical dependency between $(\boldsymbol{I} - \gamma\widehat{\boldsymbol{P}}_{\pi})^{-1}$ and $(\widehat{\boldsymbol{P}}_{\pi} - \boldsymbol{P}_{\pi})\boldsymbol{V}^{\pi}$, a natural strategy is to control these two terms separately and then to combine bounds; see [1, Lemma 5]. This simple approach, however, leads to sub-optimal bounds. To refine analysis, we further expand (15) as

$$(15) = \gamma\left(\boldsymbol{I} - \gamma\boldsymbol{P}_{\pi}\right)^{-1}\left(\widehat{\boldsymbol{P}}_{\pi} - \boldsymbol{P}_{\pi}\right)\boldsymbol{V}^{\pi} + \gamma\left\{\left(\boldsymbol{I} - \gamma\widehat{\boldsymbol{P}}_{\pi}\right)^{-1} - \left(\boldsymbol{I} - \gamma\boldsymbol{P}_{\pi}\right)^{-1}\right\}\left(\widehat{\boldsymbol{P}}_{\pi} - \boldsymbol{P}_{\pi}\right)\boldsymbol{V}^{\pi}, \tag{16}$$

which can be viewed as a "second-order" expansion. The advantage is that the first term in (16) becomes easier to cope with than its counterpart (15), owing to the independence between $(\boldsymbol{I}-\gamma\boldsymbol{P}_{\pi})^{-1}$ and $(\widehat{\boldsymbol{P}}_{\pi} - \boldsymbol{P}_{\pi})\boldsymbol{V}^{\pi}$. However, the second term in (16) remains difficult to control optimally, and it turns out that this issue can be remedied by resorting to further expansion. We shall continue to expand it to some log order, which eventually allows for optimal control of the estimation error.

**Analysis: model-based planning.** We then move on to outline the key ideas for model-based planning (Theorem 1). We shall start with the unperturbed version, which will elucidate the rationale of reward perturbation. First, we make note of the following elementary decomposition:

$$\boldsymbol{V}^{\star} - \boldsymbol{V}^{\widehat{\pi}^{\star}} = \left(\widehat{\boldsymbol{V}}^{\widehat{\pi}^{\star}} - \boldsymbol{V}^{\widehat{\pi}^{\star}}\right) + \left(\widehat{\boldsymbol{V}}^{\pi^{\star}} - \widehat{\boldsymbol{V}}^{\widehat{\pi}^{\star}}\right) + \left(\boldsymbol{V}^{\star} - \widehat{\boldsymbol{V}}^{\pi^{\star}}\right)$$

$$\leq \left(\widehat{\boldsymbol{V}}^{\widehat{\pi}^{\star}} - \boldsymbol{V}^{\widehat{\pi}^{\star}}\right) + \left(\boldsymbol{V}^{\star} - \widehat{\boldsymbol{V}}^{\pi^{\star}}\right), \tag{17}$$

which uses $\widehat{\boldsymbol{V}}^{\pi^{\star}} \leq \widehat{\boldsymbol{V}}^{\widehat{\pi}^{\star}}$ due to the optimality of $\widehat{\pi}^{\star}$. This leaves us with two terms to control.

*Step 1: bounding $\|\boldsymbol{V}^{\pi^{\star}} - \widehat{\boldsymbol{V}}^{\pi^{\star}}\|_{\infty}$.* Given that $\pi^{\star}$ is a fixed policy independent of the data, we can carry out this step using our result for model-based policy evaluation (cf. Lemma 1). Specifically, taking $\pi = \pi^{\star}$ in Lemma 1 yields that, with probability at least $1 - \delta$,

$$\left\|\widehat{\boldsymbol{V}}^{\pi^{\star}} - \boldsymbol{V}^{\pi^{\star}}\right\|_{\infty} \leq 6\sqrt{\frac{2\log\left(\frac{4|\mathcal{S}|\log\frac{e}{1-\gamma}}{\delta}\right)}{N(1-\gamma)^3}}. \tag{18}$$

*Step 2: bounding $\|\widehat{\boldsymbol{V}}^{\widehat{\pi}^{\star}} - \boldsymbol{V}^{\widehat{\pi}^{\star}}\|_{\infty}$.* Extending the result in Step 1 to $\|\widehat{\boldsymbol{V}}^{\widehat{\pi}^{\star}} - \boldsymbol{V}^{\widehat{\pi}^{\star}}\|_{\infty}$ is considerably more challenging, primarily due to the complicated statistical dependency between $(\boldsymbol{V}^{\widehat{\pi}^{\star}}, \widehat{\boldsymbol{V}}^{\widehat{\pi}^{\star}})$ and $\widehat{\boldsymbol{P}}$. The recent work [1] developed a clever "leave-one-out" type argument by constructing some auxiliary state-absorbing MDPs to decouple the statistical dependency. However, their argument requires $\varepsilon < 1/\sqrt{1-\gamma}$ and falls short of accommodating the full $\varepsilon$-range. To address this challenge, our analysis consists of the following steps, all of which require new techniques beyond [1].

• *Decoupling statistical dependency between $\widehat{\pi}^\star$ and $\widehat{P}$.* Instead of decoupling the statistical dependency between $\widehat{V}^{\widehat{\pi}^\star}$ and $\widehat{P}$ as in [1], we attempt to decouple the dependency between the policy $\widehat{\pi}^\star$ and $\widehat{P}$. If this could be achieved, then the proof strategy proposed for fixed policies would be applicable. The key ingredient of this step lies in the construction of auxiliary state-action-absorbing MDPs (motivated by [1]), as well as the development of optimal value function estimation guarantees under certain Berstein-type conditions (rather than independence assumptions). These ingredients together allow us to get hold of $\|V^{\widehat{\pi}^\star} - \widehat{V}^{\widehat{\pi}^\star}\|_\infty$, provided a certain separation condition holds.

• *Tie-breaking via reward perturbation.* An inevitable shortcoming of the above-mentioned approach is that it relies crucially on the separability of $\widehat{\pi}^\star$ from other policies; in other words, the proof might fail if $\widehat{\pi}^\star$ is non-unique or not sufficiently distinguishable from others. Consequently, it remains to ensure that the optimal policy $\widehat{\pi}^\star$ stands out from all the rest for all MDPs of interest. We show that this can be guaranteed by randomly perturbing the reward function (so as to break the ties).

## 4.2 More detailed descriptions of Step 2

We shall now flesh out the key ideas mentioned above for Step 2, consisting of three sub-steps.

*Step 2.1: value function estimation for a policy obeying Bernstein-type conditions.* We start by pointing out that Lemma 1 can be generalized far beyond the family of fixed policies (namely, those independent of $\widehat{P}$), as long as a certain Bernstein-type condition — which captures certain weak statistical dependency and will be formalized in (20) — is satisfied. To make it precise, we introduce:

$$
\begin{aligned}
\boldsymbol{r}^{(0)} &:= \boldsymbol{r}_\pi, & \boldsymbol{V}^{(0)} &:= (\boldsymbol{I} - \gamma \boldsymbol{P}_\pi)^{-1} \boldsymbol{r}^{(0)}, \\
\boldsymbol{r}^{(l)} &:= \sqrt{\mathsf{Var}_{\boldsymbol{P}_\pi}\big[\boldsymbol{V}^{(l-1)}\big]}, & \boldsymbol{V}^{(l)} &:= (\boldsymbol{I} - \gamma \boldsymbol{P}_\pi)^{-1} \boldsymbol{r}^{(l)}, & l \geq 1.
\end{aligned}
\tag{19}
$$

Our generalization of Lemma 1 is as follows, which does *not* require independence between $\pi$ and $\widehat{P}$.

**Lemma 2.** *Suppose that there exists some quantity $\beta_1 > 0$ such that $\{\boldsymbol{V}^{(l)}\}$ (cf. (19)) obeys*

$$
\left| (\widehat{\boldsymbol{P}}_\pi - \boldsymbol{P}_\pi) \boldsymbol{V}^{(l)} \right| \leq \sqrt{\frac{\beta_1}{N}} \sqrt{\mathsf{Var}_{\boldsymbol{P}_\pi}\big[\boldsymbol{V}^{(l)}\big]} + \frac{\beta_1 \|\boldsymbol{V}^{(l)}\|_\infty}{N} \boldsymbol{1}, \qquad \forall 0 \leq l \leq \log\left(\frac{e}{1-\gamma}\right). \tag{20}
$$

*Suppose $N > \frac{16e^2}{1-\gamma}\beta_1$. Then one has $\left\|\widehat{\boldsymbol{V}}^\pi - \boldsymbol{V}^\pi\right\|_\infty \leq \frac{6}{1-\gamma}\sqrt{\frac{\beta_1}{N(1-\gamma)}}$.*

*Step 2.2: decoupling statistical dependency via $(s,a)$-absorbing MDPs.* We are now positioned to show how to decouple the complicated statistical dependency between the optimal policy $\widehat{\pi}^\star$ and $\widehat{V}^\star$. Towards this, we resort to a leave-one-row-out argument, largely motivated by the novel construction in [1]. In comparison to [1] that introduces state-absorbing MDPs (so that a state $s$ is absorbing regardless of the subsequent actions), our construction is a set of state-action-absorbing MDPs, in which a state $s$ is absorbing only when a designated action $a$ is always executed at the state $s$. Specifically, for each state-action pair $(s,a)$ and each scalar $u$ with $|u| \leq 1/(1-\gamma)$, construct an auxiliary MDP $\widehat{\mathcal{M}}_{s,a,u}$ — it is identical to the original $\widehat{\mathcal{M}}$ except that it is absorbing in state $s$ if we always choose action $a$ in state $s$; namely, its probability transition kernel obeys

$$
\begin{aligned}
P_{\widehat{\mathcal{M}}_{s,a,u}}(s \mid s, a) &= 1, \\
P_{\widehat{\mathcal{M}}_{s,a,u}}(s' \mid s, a) &= 0, & \text{for all } s' \neq s, \\
P_{\widehat{\mathcal{M}}_{s,a,u}}(\cdot \mid s', a') &= P_{\widehat{\mathcal{M}}}(\cdot \mid s', a'), & \text{for all } (s', a') \neq (s, a),
\end{aligned}
\tag{21}
$$

where $P_{\widehat{\mathcal{M}}}$ is the transition kernel of $\widehat{\mathcal{M}}$. The instant reward received at $(s,a)$ in $\widehat{\mathcal{M}}_{s,a,u}$ is set to be $u$, while all other rewards stay unchanged. Let $\widehat{\pi}^\star_{s,a,u}$ be the optimal policy in $\widehat{\mathcal{M}}_{s,a,u}$. The main advantage is: for any fixed $u$, the MDP $\widehat{\mathcal{M}}_{s,a,u}$ is statistically independent of the $(s,a)$-th row of $\widehat{P}$.

It turns out that we can **represent $\widehat{\pi}^\star$ via a few policies independent of $\widehat{P}_{s,a}$**, as long as the original $\widehat{Q}^\star$ satisfies a sort of separation condition (which indicates that there is no tie when it comes to the optimal policy). More precisely, given any $0 < \omega < 1$ we define the separation condition as

$$
\mathcal{B}_\omega := \left\{ \widehat{Q}^\star(s, \widehat{\pi}^\star(s)) - \max_{a: a \neq \widehat{\pi}^\star(s)} \widehat{Q}^\star(s, a) \geq \omega \ \text{ for all } s \in \mathcal{S} \right\}. \tag{22}
$$

Our result is stated below. Here, $\mathcal{N}_\epsilon$ is an $\epsilon$-net of the interval $\left[-\frac{1}{1-\gamma}, \frac{1}{1-\gamma}\right]$.

**Lemma 3.** *Consider any $\omega > 0$, and suppose the event $\mathcal{B}_\omega$ (cf. (22)) holds. Then for any pair $(s, a) \in \mathcal{S} \times \mathcal{A}$, there exists a point $u_0 \in \mathcal{N}_{(1-\gamma)\omega/4}$, such that*

$$\widehat{\pi}^\star = \widehat{\pi}^\star_{s,a,u_0}. \tag{23}$$

*Step 2.3: a tie-breaking argument.* Unfortunately, the separation condition specified in (22) does not always hold. In order to accommodate all possible MDPs of interest without imposing such a special separation condition, we put forward a perturbation argument, allowing one to generate a new MDP that *(i)* satisfies the separation condition, and that *(ii)* is sufficiently close to the original MDP. Specifically, we aim to show that: by randomly perturbing the reward function, we can "break the tie" in the Q-function and ensure sufficient separation of Q-values associated with different actions. To formalize it, we need to introduce additional notation. Denote by $\pi_p^\star$ the optimal policy of the MDP $\mathcal{M}_p = (\mathcal{S}, \mathcal{A}, \boldsymbol{P}, r_p, \gamma)$, and $Q_p^\star$ its optimal state-action value function. We can define $\widehat{Q}_p^\star$ and $\widehat{\pi}_p^\star$ analogously for the MDP $\widehat{\mathcal{M}}_p = (\mathcal{S}, \mathcal{A}, \widehat{\boldsymbol{P}}, r_p, \gamma)$. Our result is phrased as follows.

**Lemma 4.** *Consider the perturbed reward defined in (6). With probability at least $1 - \delta$,*

$$\forall (s,a) \in \mathcal{S} \times \mathcal{A} \text{ with } a \neq \pi_p^\star(s): \qquad Q_p^\star(s, \pi_p^\star(s)) - Q_p^\star(s, a) > \frac{\xi\delta(1-\gamma)}{4|\mathcal{S}||\mathcal{A}|^2}. \tag{24}$$

*This result holds unchanged if $(Q_p^\star, \pi_p^\star)$ is replaced by $(\widehat{Q}_p^\star, \widehat{\pi}_p^\star)$.*

Lemma 4 reveals that at least a polynomially small degree of separation ($\omega = \frac{\xi\delta(1-\gamma)}{4|\mathcal{S}||\mathcal{A}|^2}$) arises upon random perturbation (with size $\xi$) of the reward function. As we shall see, this level of separation suffices for our purpose, which together with Steps 2.1-2.2 allows us to justify our main theorems.

## 5  Concluding remarks

This paper uncovers that a *perturbed* model-based planning algorithm enables minimax sample complexity in a generative model, as soon as the sample size exceeds $\frac{|\mathcal{S}||\mathcal{A}|}{1-\gamma}$ (up to log factor). Compared to prior literature, our result considerably broadens the sample size range, allowing us to pin down a complete trade-off curve between sample complexities and accuracy. The present work opens up several directions for future investigation. For instance, a natural question arises regarding the necessity of perturbation — is it possible to achieve optimality by directly performing planning on the original empirical MDP without perturbation? Our analysis framework might also shed light on how to improve sample complexity guarantees for model-free algorithms and finite-horizon MDPs.

## Broader Impact

This work is a theoretical contribution to characterize the minimax optimality of model-based reinforcement learning. The insights from the proposed algorithm can potentially be leveraged in various reinforcement learning tasks in the future.

## Acknowledgments and Disclosure of Funding

G. Li and Y. Gu are supported in part by the grant NSFC-61971266. Y. Wei is supported in part by the grants NSF CCF-2007911 and DMS-2015447. Y. Chi is supported in part by the grants ONR N00014-18-1-2142 and N00014-19-1-2404, ARO W911NF-18-1-0303, and NSF CCF-1806154 and CCF-2007911. Y. Chen is supported in part by the grants AFOSR YIP award FA9550-19-1-0030, ONR N00014-19-1-2120, ARO YIP award W911NF-20-1-0097, ARO W911NF-18-1-0303, NSF CCF-1907661, DMS-2014279 and IIS-1900140. We thank Qiwen Cui for pointing out an issue in the proof of Lemma 4 in an early version of this paper, and thank Shicong Cen, Chen Cheng and Cong Ma for numerous discussions.

## Footnotes

[1] Here, the "model" refers to the probability transition kernel and the rewards of the MDP taken collectively.

[2]Perturbation is only invoked when running the planning algorithms and does not require new samples.

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
