[Reviews · NeurIPS 2020]

Review 1

Summary and Contributions: The paper looks at the sample complexity of reinforcement learning using a generative model by building an empirical model. Previous works (especially AKY20 in the model based context) have shown that model-based planning achieves minimax sample complexity upto logarithmic factors. However, these results come with a condition on the accuracy range on \epsilon or effectively a minimum sample size for the bounds to kick in. The best results before this paper [AKY20] showed the minimax optimality for \epsilon in range (0,1/\sqrt{1-\gamma}]. Thus, the sample size has to be at least O(1/(1-\gamma)^2) per state-action pair. The results in this paper show that if one perturbs the reward during planning in the empirical MDP, the sample complexity if minimax optimal and works in the sample sample range of O([1/(1-\gamma), 1/(1-\gamma)^2]). As only the rewards are perturbed, there is no significant change in the algorithmic component and new techniques and refined analysis leads to the improved result 'breaking the sample barrier'.

Strengths: The paper is very well written and shows an improved sample complexity analysis for model-based planning with a generative model. The analysis is novel and uses new techniques to remove the aforementioned sample barrier. The key ideas in the analysis as discussed in the paper are: 1. Expand the term \hat{V}-V into variance of value functions with rewards being iterated into higher order expansions. This leads to a much improved error decomposition than a first order expansion in eq 59 in the appendix. 2. Bounding the inf norm of expected square root of variance of the value function as the inf norm of value function and not rewards, [Lemma B.1] 3. Modified way to get statistical independence: builds on AKY20 to get another way to decouple \hat{\pi}^* from \hat{P}. 4. The statistical independence step above is requires a gap assumption over Q values which won't hold in general in the empirical MDP. As such, the authors come up with a perturbation approach which guarantees that with high probability and does not change the value function much. As such it can be clearly seen that there is significant technical novelty and it results in improved sample complexity analysis. UPDATE: Thanks for the responses. I would maintain my accept rating.

Weaknesses: The planning algorithm requires some perturbation on the rewards of the empirical MDP. It clearly solves the requirement for the analysis but the maybe a refined analysis which can utilize such gap structures in the true MDP can be used. It would be interesting to see if such gaps naturally occur with same sample sizes if the gap exists in the true MDP.

Correctness: I went through the proof in the appendix briefly and to that extent the proof looks correct.

Clarity: The paper is quite well written and appropriately discusses relevant work with proper attribution of the used techniques.

Relation to Prior Work: See above.

Reproducibility: Yes

Additional Feedback:


Review 2

Summary and Contributions: This work considers the sample complexity of obtaining an epsilon-optimal policy in a discounted MDP, given only access to a generative model. The bound matches the state of the art and the contribution is related to the validity under a lower number of samples as compared to previous works. The main contribution is that the bound is useful as soon as the sample size exceeds |S||A|/(1−\gamma) (modulo some log factor). As pointed out in the paper, this is particularly interesting because this covers the full range of sample sizes that should guarantee one to find a policy strictly better than a random guess.

Strengths: The paper allows to broaden the minimax optimality in the range of \epsilon up to 1/(1-gamma)^(1/2) (modulo some log factor), which is important because it allows accommodating the entire range of sample sizes (for which a policy strictly better than a random guess can be obtained). The paper introduces model-based planning in an MDP with perturbed rewards for the proof. An intermediate result of interest is about model-based policy evaluation as soon as the number of samples is of the order of 1/(1-\gamma).

Weaknesses: In addition to the future directions mentioned in the paper, empirical illustrations of the minimax optimality for different sample sizes might be an added value to the paper.

Correctness: As far as I can tell, the method is correct and the explanations are sound.

Clarity: The paper is well-written. All notations are carefully defined. The main theorems are clearly discussed by going from the main ideas down to the detailed explanations further in the paper and in the appendix. The main steps of the analysis are clearly identified. - line 190, the dimensions of V* might need to be defined for consistency since it is defined for Q*.

Relation to Prior Work: The prior work is well-discussed. The contribution is well identified both in the introductory parts of the paper and in the key ideas of the proofs. The main result concerning the validity with small-to-moderate sample sizes of an epsilon-optimal policy is well-explained. An additional result concerning the accuracy of the model-based policy evaluation (in the empirical MDP under a fixed policy pi) under limited sample size is also provided.

Reproducibility: Yes

Additional Feedback: - In Remark 1 below theorem 3, it is mentioned that \alpha can be any number > 1. Why is 5 chosen in the formulation of Theorem 1 then (and in line 231 of the appendix)? - It might be of interest to explain in more details and earlier in the text where/why the minimum sample size of |S||A|/(1−\gamma)^2 and epsilon<1/(1-gamma)^(1/2) appeared in the previous related work [1]. --- UPDATE: I've read the other reviews and the rebuttal. My main questions and minor remarks have been answered in the rebuttal. I maintain my score of 7.


Review 3

Summary and Contributions: The authors introduce a “perturbed” model-based planning algorithm for an epsilon-optimal policy with sample complexity |S||A|/[(1-gamma)^3 epsilon^2] log [ |S||A|/[(1-gamma) epsilon]], improving upon previous works by supporting the full accuracy level range of epsilon (0, 1/(1-gamma)) instead of only (0, 1/(1-gamma)^2). Therefore, the algorithm supports minimax optimality of model-based reinforcement learning as soon as the sample size exceeds |S||A|/(1-gamma) instead of |S||A|/(1-gamma)^2. The result may be of particular interest in applications with small-to-moderate sample sizes as previous bounds were not applicable in this non-asymptotic regime. The overall idea of the algorithm employs an empirical MDP with perturbed (based on a uniform distribution) rewards. The key ideas of the model-based policy evaluation results rely on writing the Bellman equation in matrix notation, and expanding the error between true value function and the model-based empirical estimate (with an empirical estimate of the transition kernel) to second order (and further). The key ideas of the model-based planning analysis consist of two ideas to ease bounding two terms in the value error. 1) decoupling statistical dependency between \hat{\pi}^* and \hat{P}, without requiring statistical independence, through introducing state-action-absorbing MDPs. 2) Tie breaking through reward perturbation.

Strengths: Good and clear overview of related approaches in Table1 with sample size range, complexity, and epsilon-range. The contribution is outlined very clearly and nicely embedded into referrals to related works. The contribution is interesting and fills a necessary gap. Practical implications will be appreciated by the broader NeurIPS community.

Weaknesses: I feel liked the motivation for the perturbed rewards is unclear until very late in the paper. Giving an intuition earlier on would be very helpful. I understand that the structure follows more the direction of related works and nicely embeds it into the ideas of other approaches (such as [1]) and a strong overview of the related work is given and the contribution is outlined clearly. Nonetheless I would have preferred to not have to wait until the last page (and then delving into the appendix) to understand why the perturbation was chosen. It would be nice if the appendix could be structured more as a supplement rather than a repetition of the text. Right now it seems like excerpts from the appendix have been placed into the main document instead of the appendix supporting the main document. I understand that this is a theoretical contribution, but I would have liked to see some form of basic empirical evaluation.

Correctness: The paper builds on ideas from [1], namely constructing utilizing auxiliary state-absorbing MDPs to decouple state dependency. Employing auxiliary state-action absorbing MDPs instead to decouple \hat{\pi}^* and \hat{P} is a quite clever idea. Likewise, breaking ties through random reward perturbation is a valid and interesting approach.

Clarity: Overall, the paper is written clearly and well-motivated. As mentioned above, I would have preferred a slightly different structure where the main contributions, specifically how they work, are mentioned earlier.

Relation to Prior Work: Yes!

Reproducibility: Yes

Additional Feedback: While this work is in theory, I would have hoped for at least some statement that discusses the impact on society through automation, or how this specific contribution (in the low sample regime) may aid the development of new technologies in more detail. (This is very minor though) ----------------After rebuttal------------------------------------------------------------------------- I believe my concerns have been addressed and I am happy to see some quantitative results added to the paper.


Review 4

Summary and Contributions: The paper analyzes the sample complexity of discounted MDPs with access to samples from the true MDP. Prior work has identified scaling properties for the number of samples required in terms of the state action sizes to reach a given epsilon error threshold for the estimates. However, prior analysis does not cover the range where this epsilon error is high (up to 1/1-gamma, where gamma is the horizon). The current paper focuses on this high error range and derives a result where the effective scaling, when viewed in terms of the horizon gamma, can be better in this high range of epsilon. This is accomplished by adding noise to the deterministic reward function samples.

Strengths: One of the strengths of this paper is a novel focus on the particular range of epsilon which could result in a lower overall complexity in terms of the horizon gamma. The observation that the true sample complexity in terms of gamma can be lower than prior stated results under certain conditions where we tolerate high error seems novel.

Weaknesses: The significance of "breaking the barrier" is somewhat suspicious since it appears to be relevant only when there is a lower bound assumption on the accuracy epsilon, which is a bit strange since we want the accuracy to be high, so the error epsilon to be low. In particular, it doesn't appear to improve on previous results if we make epsilon a constant, for example. EDIT: Thank you to the authors for your response. Here is a bit more explanation of my concern. My comment was inspired by thinking about what are the conditions under which the new bound derived by the authors is actually a strict improvement over the previous bound. As the authors re-state in their response, the previous bound is a max of two terms, one of which depends on epsilon and the other on gamma. As the authors state, they removed the gamma dependent term from the previous bound. As I pointed out in the original review, this is an improvement only when the gamma-dependent term is larger than the epsilon dependent term in the prior bound. In other words, unless epsilon is large (i.e. the required approximation guarantee required is loose), the new bound is not better than the old one. However, I do understand now that there is a range of epsilon when gamma is large when the complexity is an improvement, so I am revising my score upward. The perturbation condition for the noise requires a variance that scales as epsilon/S^alpha A^alpha, which means that for larger state-action pairs, the perturbation is converging to zero.

Correctness: I do have some concerns around the scaling theorems. The claims are made in a way that the allowable accuracy range is expanded, however, the improved complexity claim e.g. Eq (10) seems conditional on a lower bound for epsilon and hence is not universal. In this sense, I think the claims are not quite fully accurate.

Clarity: The original model assumes a deterministic reward function. The authors propose adding noise to this true reward function and then using those samples in the estimate. It is not clear how the perturbed reward samples are combined with the transition samples in the dataset. In particular, are the perturbed reward samples counted towards the sample complexity limit? The role of the perturbation is also not clear, not withstanding remark 2. Intuitively, it seems implausible that adding noise to a known deterministic reward function is a critical part of the derived result.

Relation to Prior Work: Yes, the discussion of prior work seems fairly thorough and well explained.

Reproducibility: Yes

Additional Feedback: In Remark 1 (Line 123), the authors claim that the theorem remains unchanged for any alpha \ge 1. Why is it stated for alpha = 5 in this case given that a larger alpha requires strictly less noise. Are there missing state-action dependencies in Eq (10)?

[Author Response · NeurIPS 2020]

We thank all reviewers for very helpful comments. This letter addresses several major questions raised by the reviewers.

**A common question: numerical evaluation.** We will add a series of nu-
merical experiments to demonstrate the minimax optimality of the model-
based approach studied herein. Fig. 1 depicts some numerical plots of this
kind. Here, we adopt a (least favorable) example designed in the minimax
bound in Azar et al. [2] (and also Wainwright [37]), which primarily fo-
cuses on the effect of the discount complexity $\frac{1}{1-\gamma}$ on the sample complex-
ity. More specifically, consider the MDP $(\mathcal{S}, \mathcal{A}, P, r, \gamma)$, where $\mathcal{S} = \{0, 1\}$,
$\mathcal{A}_0 = \{0\}, \mathcal{A}_1 = \{0, 1\}, P(0|0, 0) = 1, P(1|1, 0) = p, P(1|1, 1) = q$ and
$r(0, 0) = 0, r(1, 0) = r(1, 1) = 1$. According to Azar et al., the quantities $p$
and $q$ are taken to be $p = \gamma + \frac{2\gamma(1-\gamma)^2\varepsilon}{(1+\gamma)^2}$ and $q = \gamma - \frac{2\gamma(1-\gamma)^2\varepsilon}{(1+\gamma)^2}$. As illustrated
in Fig. 1, the numerical sample complexity per state-action pair $N$ scales
on the order of $\frac{1}{(1-\gamma)^3\varepsilon^2}$ for varying choices of accuracy level $\varepsilon$, which is
consistent with our theory.

Figure 1: Numerical experiments.

**Specific questions by Reviewer 1:** 1. *Necessesity of reward perturbation.*
Indeed, reward perturbation is introduced merely to facilitate analysis. Given
that we recommend extremely small perturbations, the difference between the
perturbed and original MDPs is almost unnoticeable in the experiments. While
our current analysis does not work without reward perturbation, it would be
important to see whether this can be removed with more refined analysis.

**Specific questions by Reviewer 2:** 1. *Empirical illustration of minimax optimality.* We have conducted some
illustrative numerical experiments to address this comment; see the response above for "Numerical evaluation".

2. *Choice of $\alpha$.* Indeed, our theory and analysis hold for any constant $\alpha > 1$. The current paper picks the specific choice
$\alpha = 5$ only to convey that the perturbation $\frac{(1-\gamma)\varepsilon}{|\mathcal{S}|^\alpha|\mathcal{A}|^\alpha}$ can be very small; we shall make it more clear in the final version.

3. *Why the sample size barrier $N > 1/(1 - \gamma)^2$ appeared in Agarwal et al. [1].* Take Section 4.3 of the Arxiv version
of Agarwal et al. for example: the contraction factor $\gamma\sqrt{\frac{8\log(|\mathcal{S}||\mathcal{A}|/(1-\gamma)\delta)}{N}}\frac{1}{1-\gamma}$ needs to be smaller than 1, which
requires the sample size per state-action pair to exceed $N > 1/(1 - \gamma)^2$ (up to log factor). We shall explain it in the
final paper.

**Specific questions by Reviewer 3:** 1. *Motivation for perturbed rewards and paper structure*: Thanks for the helpful
suggestion. We will elucidate the motivation and intuition of reward perturbation earlier on in the revised paper.

2. *Empirical evaluation*: We have conducted some illustrative numerical experiments which will be added to the revised
paper; see the response above on "Numerical evaluation".

3. *Paper structure and broader impacts*: We will restructure the appendix (so as to reduce repetition) and add new
statements about potential societal impacts as suggested by the reviewer.

**Specific questions by Reviewer 4:** 1. *Constraint on accuracy level $\varepsilon$.* In fact, our result (see Theorem 1) holds for an
arbitrary choice of $\varepsilon \in (0, \frac{1}{1-\gamma}]$. This accommodates an arbitrary $\varepsilon$, and there is no lower bound on the accuracy level $\varepsilon$
in our main theorem. To be a bit more specific, our sample complexity bound reads $\frac{|\mathcal{S}||\mathcal{A}|}{(1-\gamma)^3\varepsilon^2}$, while prior work like
Agarwal et al. reads $\max\left\{\frac{|\mathcal{S}||\mathcal{A}|}{(1-\gamma)^3\varepsilon^2}, \frac{|\mathcal{S}||\mathcal{A}|}{(1-\gamma)^2}\right\}$ (up to some log factors). Here, the red text represents the sample size
barrier, and is removed in our theory. We understand from the reviewer's comment that there might be confusion in our
current discussions/remarks, and we shall rephrase them in the final paper to clarify any such confusion.

2. *The role of reward perturbation and choice of $\alpha$.* Reward perturbation is introduced mainly to break the ties (so as
to ensure uniqueness of optimal policy). Fortunately, even an extremely small level of perturbation suffices for this
purpose (as long as it is not exponentially small). Our analysis and theorems hold for any fixed constant $\alpha > 1$. Here,
we pick $\alpha = 5$ only to emphasize that a fairly small level of reward perturbation suffices for our analysis to work (more
specifically, a fairly small level of perturbation suffices in breaking the ties). This will be made clear in the final paper.

3. *About perturbed reward samples.* Here, reward perturbation is only enforced when running the planning algorithms
on empirical MDPs (basically, we collect the true reward samples $r$, but use $r_{\mathrm{p}} = r + \zeta$ in the algorithm). In other
words, it is not necessary to collect new samples for this purpose; we shall clarify this point in the revision.

4. *Missing state-action dependency in Eq. (10).* Here, $N$ is defined as the sample size *per state-action pair* (so that the
total sample size should be $|\mathcal{S}||\mathcal{A}|N$. We shall make it more clear in the revision.

[Meta-Review · NeurIPS 2020]

The reviewers appreciated the efforts made by the authors in the rebuttal and updated their reviews accordingly. The reviewers are now all positive about the paper. They are aware that the improvements of the results concern specific regimes for \epsilon, \gamma, but appreciate the results on this fundamental problem. We recommend the paper for acceptance and encourage the authors to account for the reviewers’ comments when preparing the camera-ready version of the paper.